# Cohesin reconstitution and homologous recombination repair of DNA double-strand breaks in late mitosis

**Jessel Ayra Plasencia[1,2]*[†‡], Sara Medina-Suárez[1,2][†],
Esperanza Hernández-Carralero[1], Jonay García-Luis[2], Lorraine S Symington[3],
Félix Machín[1,2,4]***

[1]Hospital Universitario Nuestra Señora de Candelaria, Instituto de Investigación Sanitaria de Canarias (IISC), Santa Cruz de Tenerife, Spain; [2]Instituto of Tecnologías Biomédicas, Universidad de La Laguna, San Cristóbal de La Laguna, Spain; [3]Department of Microbiology and Immunology, Columbia University Medical Center, New York, United States; [4]Facultad de Ciencias de la Salud, Universidad Fernando Pessoa Canarias, Las Palmas de Gran Canaria, Spain

**\*For correspondence:**
jessel.ayra.plasencia@gmail.com
(JAP);
fmachin@fciisc.es (FM)

[†]These authors contributed
equally to this work

**Present address:** [‡]Unidad
de Investigación, Hospital
Universitario de Canarias, San
Cristóbal de La Laguna, Spain

**Competing interest:** The authors
declare that no competing
interests exist.

**Reviewing Editor:** Pablo
S Aguilar, Instituto de
Fisiología Biología Molecular
y Neurociencias (IFIBYNE),
Argentina

## eLife Assessment

This study provides **convincing** evidence that homologous recombination can occur in telophase-arrested cells, independently of cohesin subunits Smc 1-3. These findings are **valuable** as they point to investigate the role of cohesins re-association with chromatin in the allelic inter-sister repair by homologous recombination.

**Abstract** The cohesin complex maintains sister chromatid cohesion from S phase to anaphase onset. Cohesin also plays roles in chromosome structure and DNA repair. In yeast, the cohesin subunit Scc1 is cleaved at anaphase onset to allow segregation in an orderly manner, although some residual cohesin subunits remain to maintain chromosome structure. Efficient DNA double-strand break (DSB) repair by homologous recombination (HR) with the sister chromatid also depends on cohesin. Here, we have examined the role of residual cohesin in DSB repair in telophase (late mitosis). We have found that Scc1 returns in telophase after DSBs and that it partially reconstitutes a chromatin-bound cohesin complex with Smc1 and an acetylated pool of Smc3 after a single HO-induced DSB at the *MAT* locus. However, this new cohesin is neither required for the HR-driven *MAT* switching nor binds to the *MAT* locus after the DSB.

## Introduction

Cell survival and genome integrity depend on the faithful segregation of chromosomes in anaphase. From the S phase until the anaphase onset, the highly conserved cohesin complex plays an essential role in the structural maintenance of sister chromatids. Cohesin is a multiprotein complex composed in yeast of Smc1, Smc3, Scc1 (also known as Mcd1), Scc3, and Pds5 subunits (*Hirano, 2000*; *Michaelis et al., 1997*; *Sjögren and Nasmyth, 2001*). At the core of the complex, Smc1, Smc3, and Scc1 form a heterotrimeric ring that embraces and holds the replicated sister chromatids together and well-aligned until reaching G2/M (*Díaz-Martínez et al., 2008*; *Koshland and Guacci, 2000*; *Laloraya et al., 2000*; *Michaelis et al., 1997*; *Strunnikov et al., 1993*). Cohesin subunits are loaded onto chromosomes in G1 by the Scc2–Scc3 complex (*Ciosk et al., 2000*). When cells enter S phase, the acetyl-transferase Eco1 acetylates Smc3, which inhibits the ATPase activity of Smc1–Smc3 heads and

prevents the opening of the Smc3–Scc1 interface (*Çamdere et al., 2015*; *Chan et al., 2012*; *Huber et al., 2016*; *Murayama and Uhlmann, 2015*; *Ström et al., 2007*; *Unal et al., 2007*). This embraces the nascent sister chromatids and establishes cohesion (*Uhlmann and Nasmyth, 1998*). Once sister chromatids are ready for segregation, the APC^Cdc20 (Anaphase Promoting Complex associated with its regulatory cofactor Cdc20) initiates anaphase by degrading securin/Pds1, so that the separase/Esp1 becomes active. Separase then cleaves the Scc1 subunit by proteolysis, releasing sister chromatids from cohesion (*Cohen-Fix et al., 1996*; *Michaelis et al., 1997*; *Nasmyth and Haering, 2009*; *Uhlmann et al., 1999*; *Yamamoto et al., 1996*). The fragmented Scc1 is then rapidly degraded (*Rao et al., 2001*), while a pool of Smc1–Smc3 dimers appears to remain through anaphase (*Renshaw et al., 2010*; *Tanaka et al., 1999*). This pool, however, becomes loose as Smc3 is deacetylated by Hos1 in anaphase (*Chan et al., 2012*; *Huber et al., 2016*; *Murayama and Uhlmann, 2015*). Interestingly, some cohesin-dependent cohesion remains at chromosome arms during anaphase, suggesting that residual cohesin is present despite separase activation (*Djeghloul et al., 2020*; *Garcia-Luis et al., 2022*; *Renshaw et al., 2010*).

In addition to its role in sister chromatid cohesion, cohesin has also been involved in chromosome structure and DNA repair. On the one hand, cohesin can hold two DNA segments within the same chromatin forming an extruded loop (*Lazar-Stefanita et al., 2017*; *Rao et al., 2017*; *Schalbetter et al., 2017*). On the other hand, DNA double-strand breaks (DSBs) produce the so-called damage-induced cohesion (DI-cohesion) (*Kim et al., 2010*). Following DNA damage, cohesin is recruited and accumulated both along 50–100 kb surrounding the DSB site and genome-wide (*Ström et al., 2004*; *Unal et al., 2004*). The cohesin recruitment creates a firm anchoring of two well-aligned sister chromatids, which in turn facilitates DSB repair by homologous recombination (HR) (*Covo et al., 2010*; *Hou et al., 2022*; *Phipps and Dubrana, 2022*). In this regard, DNA damage kinases such as Mec1 and Chk1 phosphorylate Scc1 to initiate the cohesion establishment in post-replicative cells (*Heidinger-Pauli et al., 2008*).

HR is a DSB repair mechanism that uses a homologous template for restoring the original broken DNA. HR is highly reliable when the correct template is used, which in mitotic cells is the sister chromatid. Hence, HR is the preferred repair mechanism from S to G2/M, when the sister chromatid is available in close proximity (*Langerak and Russell, 2011*; *Mathiasen and Lisby, 2014*; *Symington et al., 2014*). Before DNA replication, in G1, the alternative error-prone non-homologous end joining (NHEJ) is used instead. Cells coordinate the choice of either NHEJ or HR based on the activity of cyclin-dependent kinase (CDK). Low CDK keeps cells in G1 and favors NHEJ, whereas high CDK is present in S and G2/M and activates HR. Noticeably, there is a paradox in this link between high CDK and HR. In the cell cycle window that spans from anaphase to the telophase-to-G1 transition (we will refer to this window as late mitosis), high CDK is set to favor HR despite a sister chromatid not being in proximity (*Machín and Ayra-Plasencia, 2020*). In a previous study, we showed that sister loci can indeed move closer and coalesce, and that HR still appears important for DSB survival in late mitosis (*Ayra-Plasencia and Machín, 2019*). Considering all these observations, we wondered whether the residual cohesin is affected by DSBs in late mitosis (telophase) and whether this cohesin pool is important for HR repair.

## Results and discussion
### Scc1 becomes stable after DSBs in late mitosis

At the anaphase onset, Esp1 cleaves Scc1, opening the cohesin ring and releasing sister chromatids from cohesion. Thereafter, Scc1 needs to be translated de novo since cleaved fragments are unstable and degraded by the proteasome (*Rao et al., 2001*). However, it is known that activation of Esp1 can be blocked by DDC kinases (*Yam et al., 2020*). In addition, DDC kinases also phosphorylate Scc1 to facilitate DI-cohesion at a post-replicative stage (*Heidinger-Pauli et al., 2008*). Thus, it is feasible that DSB generation in late mitosis can stabilize de novo Scc1 as well as rendering it cohesive. Noteworthy, Smc1 and Smc3 subunits remain attached to chromatin after segregation (*Garcia-Luis et al., 2022*; *Renshaw et al., 2010*; *Tanaka et al., 1999*).

To assess how cohesin subunits behave after DSBs in late mitosis, *cdc15-2* strains that bear Scc1 tagged with myc epitopes and Smc1/3 tagged with HA epitopes were first blocked in telophase at 34°C for 3 hr (Tel) (*Figure 1A*). The *cdc15-2* allele encodes a conditionally thermosensitive Cdc15

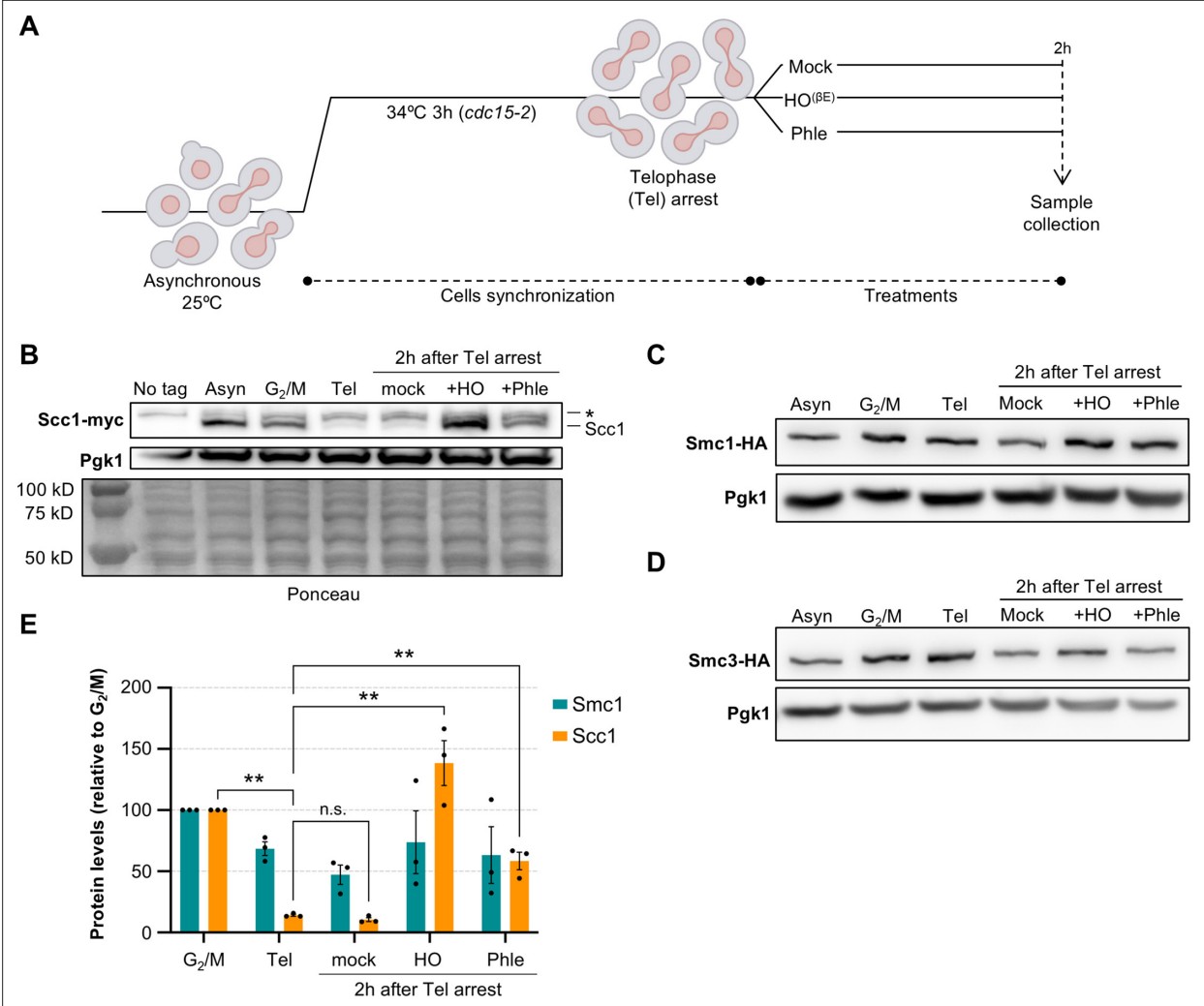

**Figure 1.** Scc1 returns after double-strand breaks (DSBs) in late mitosis. (**A**) Schematic of the experimental procedure. Cells in logarithmic growth phase were first arrested in telophase (Tel) by incubating them at 34°C for 3 hr (strains bear the *cdc15-2* allele). Then, the culture was divided into three subcultures. One served as a mock control, and the other two were treated to generate DSBs, one with β-estradiol (to express the HO endonuclease) and the other with phleomycin. The strain is unable to repair the HO DSB break by HR-driven gene conversion (Δ*hmr* Δ*hml*). (**B**) Western blot against Scc1-3myc. This experiment compares Scc1-3myc levels in an asynchronous, G2- and telophase-blocked cultures. The leftmost lane (No tag) is a control strain for Scc1 without the 3myc epitope tag. Pgk1 protein served as a housekeeping. Ponceau S staining is also shown as a loading control. Asyn.: asynchronous. Tel: telophase. +Phle: 10 mg·ml⁻¹ phleomycin. +HO: 2 μM β-estradiol. *: Unspecific band detected by the α-myc antibody just over the Scc1-3myc signal. (**C**) Like in (**B**) but against Smc1-6HA. (**D**) Like in (**C**) but against Smc3-6HA. (**E**) Quantification of Scc1-3myc and Smc1-6HA levels in all conditions (mean ± SEM, $n = 3$). Statistical comparisons are shown for selected pairs (**$p < 0.01$; one-way ANOVA, Tukey's post hoc).

The online version of this article includes the following source data for figure 1:

**Source data 1.** Original blots.

**Source data 2.** Uncropped blots in *Figure 1B*.

**Source data 3.** Uncropped blots in *Figure 1C, D*.

**Source data 4.** Values for plots.

kinase, which drives telophase-to-G1 transition (***Machín and Ayra-Plasencia, 2020***). These strains also harbor an inducible HO endonuclease, which cuts the HO cutting site (*HOcs*) at the *MAT* locus in chromosome III. The HO gene is under the control of a promoter system that responds to β-estradiol (*lexO4p:HO*:plus LexA-ER-B112-T) (***Gnügge and Symington, 2020***; ***Ottoz et al., 2014***). In wild-type strains, the HO DSB is repaired by HR with the intramolecular ectopic loci *HMR* and *HML*. The locus selection depends on the *MAT* allele (the *MAT*a HO DSB repairs with *HML*) and results in a gene

conversion (e.g., from *MAT*a to *MAT*α). In order to undoubtedly attribute the effects on cohesin to the DSB and not to downstream events of the repair process, we used a set of strains unable to repair the HO-mediated DSB (Δ*hml* Δ*hmr* double mutant). After the Tel arrest, the cultures were divided into three. The first subculture served as a mock control, the second was used to generate the single HO DSB (+HO), and the third was used to generate multiple random DSBs by means of the radiomimetic drug phleomycin (+phle). Noteworthy, these latter DSBs are repairable in principle. Incubation of the subcultures was prolonged at 34°C for 2 extra hours.

The presence or absence of Scc1-3myc, Smc1-6HA, and Smc3-6HA was checked by Western blot. As controls, we included asynchronous samples from the same strains, before the telophase arrest, as well as a parallel arrest at G2/M by the drug nocodazole (Nz). As expected, Smc1, Smc3, and Scc1 bands were detected in both cycling (asyn) cells and cells blocked in G2/M (*Figure 1B–D*). The Scc1 signal disappeared almost entirely when cells were arrested in telophase and remained low in the mock control 2 hr later. Strikingly, full-length Scc1 was restored after single and multiple DSBs generation (*Figure 1B, E*). Compared to the mock condition, phleomycin treatment increased Scc1 levels fivefold, reaching approximately half of the levels observed in G2/M. In the case of HO DSBs, Scc1 levels surpassed those of the mock by more than 15-fold and were even higher than in G2/M. In contrast, Smc1 and Smc3 levels remained more stable in both types of DSBs (*Figure 1B–D*).

## The cohesin complex is reconstituted and binds to chromatin after the HO DSBs in telophase

The above findings suggest that the cohesin complex could be newly formed and loaded onto the breaks, promoting DI-cohesion, which could in turn ease the recombinational events. To assess whether cohesin subunits are reassembled after DSB induction in telophase, we performed a co-immunopre-cipitation (co-IP) experiment using Smc1-6HA as the bait protein in a strain that also expresses Scc1-3myc (*Figure 2A*, *Figure 2—figure supplement 1*). In our experimental setup, in addition to the anti-HA and anti-myc antibodies, we included an antibody for the acetylated form of Smc3 at lysine residues K112 and K113 (acSmc3), which have been linked to the chromatin-bound pool of the Smc1–Smc3 dimer (*Beckouët et al., 2010*; *Minamino et al., 2015*; *Rolef Ben-Shahar et al., 2008*; *Unal et al., 2008*). In asynchronous cultures, both Scc1-3myc and acSmc3 co-immunoprecipitated with Smc1-6HA, indicating that we were able to pull down the chromatin-bound pool (acetylated Smc3) of the canonical Smc1–Smc3–Scc1 complex. A parallel co-IP using untagged Smc1 did not recover the other subunits, confirming the specificity of the interaction. As expected, Scc1 was not detected in the pull-down under the mock condition (no DSBs); however, it was recovered after generation of the HO DSB. Surprisingly, this was not the case when random DSBs were induced by phleomycin. Incidentally, acSmc3 appeared as two bands in the asynchronous culture but as a single upper band in all telophase conditions, suggesting a difference in posttranslational modifications at this cell cycle stage.

The reconstitution of the complex and the detection of the acetylated form of Smc3 prompted us to confirm whether the newly formed Scc1 was bound to chromatin, since the coexistence of two pools could not be ruled out based on the co-IP experiments alone. For instance, a soluble Smc1–Smc3–Scc1 complex could coexist with a chromatin-bound Smc1–acSmc3 dimer, which could represent up to 30% of the total Smc1–Smc3 in anaphase (*Beckouët et al., 2010*). To this end, we performed biochemical fractionation of the chromatin-bound proteins, observing an increase in all cohesin subunits in the chromatin fraction after the HO DSB (*Figure 2B*, *Figure 2—figure supplement 2*). The increase was approximately twofold for Smc1–acSmc3 and up to sevenfold for Scc1. These results confirmed that the new Scc1 binds to a reconstituted chromatin-bound cohesin complex following the HO DSB. In contrast, this chromatin recruitment was not observed after phleomycin treatment. The differences in co-IP and fractionation due to the source and nature of the DSBs are intriguing. In this experimental setup, the HO DSB is irreparable, whereas DSBs caused by phleomycin can eventually be repaired by anaphase regression and sister loci coalescence (*Ayra-Plasencia and Machín, 2019*). In addition, the HO DSB leaves clean ends that can be easily resected, while the DSBs after phleomycin are chemically modified and require extended processing.

Next, we used chromatin immunoprecipitation (ChIP) to investigate whether the HO DSB recruits more cohesin to the vicinity of the *HOcs*. This recruitment has been reported during DSBs that exert a G2/M arrest by the DDC (*Scherzer et al., 2022*; *Ström et al., 2004*). Because the level of Scc1 changes so dramatically after the HO DSB, we selected one of the subunits that does not for the ChIP.

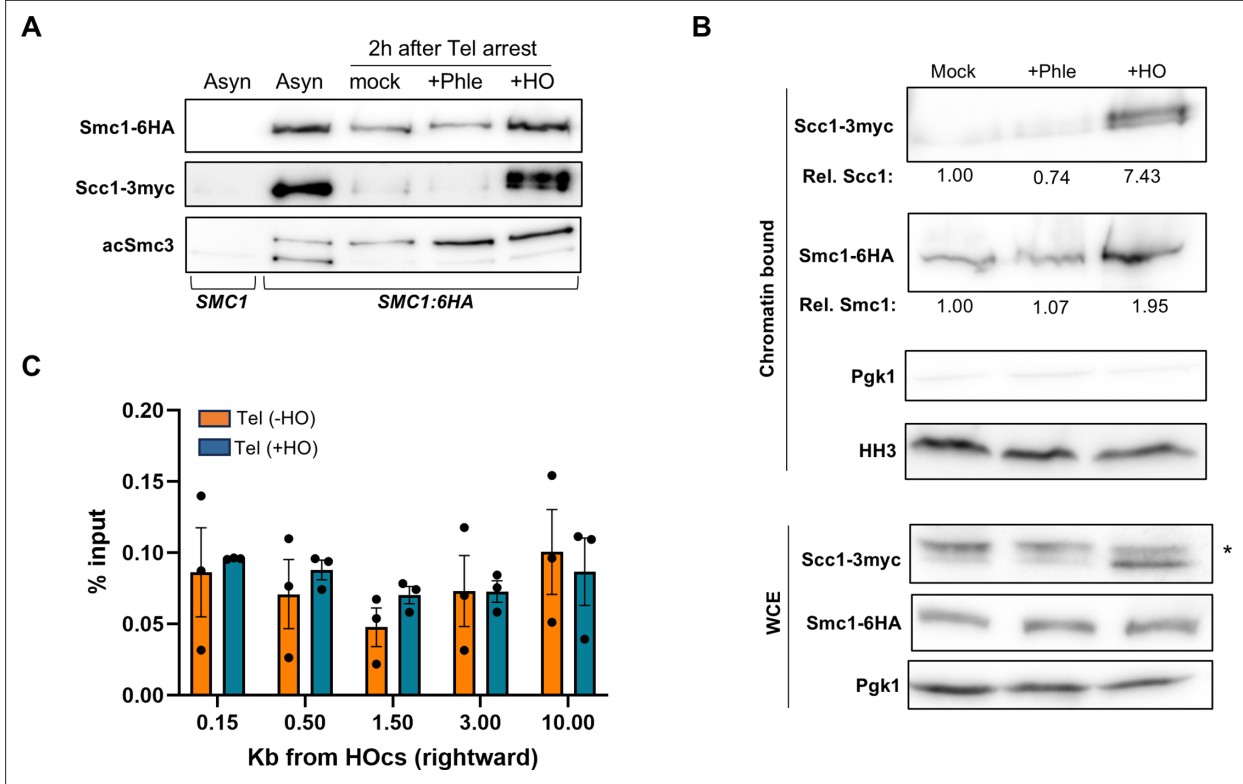

**Figure 2.** Scc1 forms a reconstituted chromatin-bound cohesin complex after the HO double-strand break (DSB) in late mitosis. Cells were treated as in *Figure 1A*. (**A**) Co-immunoprecipitation (co-IP) of the core cohesin complex. Smc1-6HA was used as the bait protein; the leftmost lane includes a control with an untagged Smc1. (**B**) Chromatin fractionation of the strain expressing Smc1-6HA and Scc1-3myc. Relative protein levels (to the mock sample) are indicated under the blots. Pgk1 and histone H3 (HH3) were included as reporters of the cytosolic and chromatin-bound fractions, respectively. The asterisk (*) indicates an unspecific band in the whole cell extract (WCE) that is absent from the chromatin-bound fraction. (**C**) Cohesin does not bind the HO DSB in telophase. Chromatin immunoprecipitation (ChIP) of Smc3-6HA with (+HO) and without (−HO) HO induction (mean ± SEM, $n = 3$).

The online version of this article includes the following source data and figure supplement(s) for figure 2:

**Source data 1.** Original blots.

**Source data 2.** Uncropped blots.

**Source data 3.** Values for plots.

**Figure supplement 1.** Co-immunoprecipitation (co-IP) of the core cohesin complex.

**Figure supplement 1—source data 1.** Original blots.

**Figure supplement 1—source data 2.** Uncropped blots.

**Figure supplement 2.** Assessment of chromatin-bound cohesin after double-strand breaks (DSBs) in late mitosis.

**Figure supplement 2—source data 1.** Original blots.

**Figure supplement 2—source data 2.** Uncropped blots.

**Figure supplement 3.** Chromatin immunoprecipitation (ChIP) of Smc3-6HA in G2/M.

**Figure supplement 3—source data 1.** Values for plots.

An Smc3-HA strain unable to repair the HO DSB (Δ*hml* Δ*hmr* double mutant) was arrested in both G2/M and telophase. We confirmed an increase in the Smc3-bound pool around the *HOcs* in G2/M after the DSB (*Figure 2—figure supplement 3*); however, such an increase was not observed in telophase (*Figure 2C*). We thus conclude that the core of the cohesin SMCs (Smc1–Smc3) is not further recruited to the HO DSB in late mitosis. This lack of local recruitment could be due to several non-exclusive scenarios, including the inability to increase the acSmc3 pools by Eco1, whose levels drop after G2/M (*Lyons and Morgan, 2011*), as well as condensin binding to chromosome arms during anaphase (*Leonard et al., 2015*), which might act as an eviction barrier for cohesin loading. However,

reassembly of the full cohesin complex at residual Smc1–acSmc3 dimers may shift the organization of the surrounding chromosome regions and impinge on the HO DSB repair in telophase.

## The HO-mediated DSB is efficiently repaired by HR in late mitosis

In a previous report, we showed from both genetic and cytological points of view that HR was still active in late mitosis (*Ayra-Plasencia and Machín, 2019*). Having observed that cohesin is reconstituted and chromatin-bound after the HO DSB in telophase, we next wondered about the functional role of the complex in this context.

We first tested whether the HO-induced DSB is efficiently resected into 3′ ssDNA tails in late mitosis. Resection is necessary to form the protruding 3′ Rad51 nucleoprotein filaments that invade the donor homologous sequence to restore the break by HR (*Peng et al., 2021*; *Symington, 2016*). To confirm and quantitate the formation of ssDNA flanking the *HOcs*, we used a qPCR approach whereby primers can only amplify a target DNA cut with the restriction enzyme *StyI* if the target has been rendered single-stranded by resection (*Figure 3—figure supplement 1*; *Gnügge et al., 2018*; *Zierhut and Diffley, 2008*). We used the Δ*hml* Δ*hmr* double mutant to hamper the normal HR flow from resection to invasion and thus boost ssDNA detection. Two different locations downstream of the *HOcs* were monitored to determine both the initial resection (0.7 kb from *HOcs*) and the kinetics further away (5.7 kb). In addition, cells arrested in either G2/M or telophase were directly compared to each other. Cells were first arrested for 3 hr, the HO was induced afterwards, and samples were taken every hour during a 4 hr time course while sustaining the HO induction (*Figure 3A*). As expected, cells in G2/M efficiently resected the *HOcs* ends. The time to reach half of the maximum resection ($t_{1/2}$) was ~1 hr at 0.7 kb and ~2.5 hr at 5.7 kb from the DSB, respectively. Resection in telophase was only slightly delayed at 0.7 kb ($t_{1/2}$ ~ 1.5 hr), whereas the delay was more pronounced at 5.7 kb ($t_{1/2}$ ~ 3.5 hr). The delay may be due to either the fact that CDK activity in late mitosis, though still high, is slightly inferior to that at G2/M or the different chromosome compaction degrees between the arrests. Together with the samples for the qPCR, samples for Western blot of Rad53 were taken. Rad53 is an effector kinase of the DDC that gets activated by hyperphosphorylation after being recruited to the 3′ ssDNA overhangs (*Branzei and Foiani, 2006*). In agreement with the resection profile, Rad53 hyperphosphorylation occurred in late mitosis, although with ~1 h delay (*Figure 3B*). This delay is also consistent with previous studies on Rad53 activation upon an unrepairable HO DSB (*Pellicioli et al., 2001*).

Once resection begins, cells are normally committed to recombine because long ssDNA fragments do not work as efficient substrates for NHEJ. In our haploid strain, HR at the *MAT* locus after the HO-driven DSB implies the gene conversion from the *MAT*a to the *MAT*α allele, whereas repair by NHEJ results in reconstitution of the *MAT*a allele (*Figure 3C*; *Yamaguchi and Haber, 2021*). This gene conversion can be detected by Southern blot because of a restriction fragment length polymorphism for *StyI*. Cells that are in G1 can barely repair the HO DSB, whereas those in G2/M do so through HR (*Ira et al., 2004*). To check whether HR was efficient at the HO DSB in late mitosis, we performed an experimental setup similar to the one shown above, but with two important differences. On the one hand, the strain retains the *HML* locus, enabling HR if engaged, and on the other hand, the HO endonuclease was transiently induced by 1 h β-estradiol pulse, allowing time for the complete repair of the induced DSB. Samples were taken at the time of the arrest, after the pulse, and all through the repair window. We confirmed by Western blot that HO (HO tagged with the Flag epitope) is produced after the pulse and is rapidly degraded afterwards (*Figure 3D*, HO-Flag blot). Rad53 hyperphosphorylation followed the ~1 hr delay shown above and became prominent by the time the DSB inducer had been removed (*Figure 3D*, Rad53 blot), suggesting extensive resection even in this repairable setup. The Southern blot showed that (1) most *MAT*a locus in the cell population has been efficiently cut after the HO pulse (*Figure 3E, F*; decrease of the *MAT*a band intensity and rise of that of the cut *HOcs*); and (2) most DSBs are repaired afterwards exclusively through HR (*Figure 3E, F*; drop of the HO cut band intensity and rise of that of the *MAT*α). No signs of DSB repair by NHEJ were observed (the remaining *MAT*a band stays constant throughout). This was further confirmed by measuring the repair kinetics of a derivative strain deficient in NHEJ (*yku70Δ*; *Figure 3—figure supplement 2A–C*), and is consistent with our previous studies with a HR-deficient *rad52Δ* strain (*Medina-Suárez et al., 2024*). Interestingly, HR kinetics were not altered in cells unable to activate the DDC (*rad9Δ*; *Figure 3—figure supplement 2D–F*) or to tether the DSB ends (*mre11Δ*; *Figure 3—figure supplement 2G–I*). These results are also consistent with previous studies in G2/M

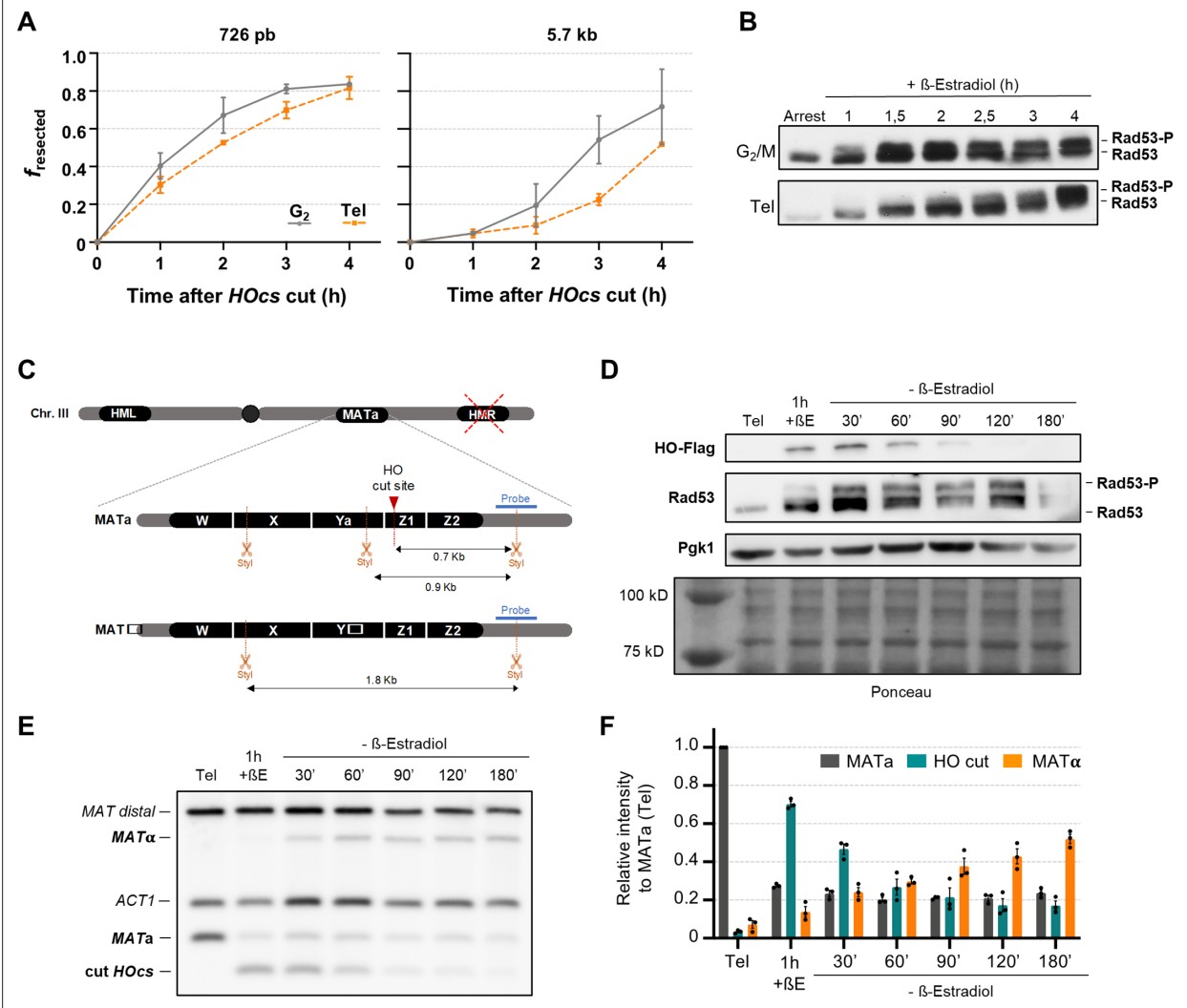

**Figure 3.** Homologous recombination (HR) repairs the HO double-strand break (DSB) in telophase. (**A, B**) Resection in late mitosis is almost as efficient as in G2/M. Cells were arrested in G2/M or telophase (Tel), and then the HO DSB was generated by adding β-estradiol. Samples were taken at the indicated times to monitor the kinetics of resection. This strain is unable to repair the HO DSB break by HR-driven gene conversion (Δhmr Δhml). (**A**) Charts depicting the resection kinetics for two different amplicons located at 726 bp and 5.7 kb downstream the *Hocs* break (mean ± SEM, n = 3; $f_{resected}$ is the proportion of resected DNA). (**B**) Representative Western blots against Rad53 to follow the sensing of DNA damage detection in G2/M versus late mitosis. (**C–F**) Yeast cells use HR to repair the HO DSB in late mitosis. Cells were first blocked in telophase (Tel). Then, the HO DSB was generated by adding β-estradiol. After 1 hr, the β-estradiol was washed away, and samples were taken to monitor the repair for 3 hr. The strain can repair the HO DSB break by HR-driven gene conversion (Δhmr HML). (**C**) Schematic of the fragments obtained after a StyI digestion for both *MAT*a and *MAT*α sequences. The probe to detect the fragments by Southern blot is in blue. When the *MAT*a locus is intact, the digestion gives rise to a 0.9 kb fragment. The HO cutting site (*Hocs*) is located within the StyI-digested *MAT*a locus. Thus, the HO-driven DSB shortens the fragment to 0.7 kb. HR leads to a gene conversion to *MAT*α, which results in the loss of a *StyI* restriction site and a new fragment of 1.8 kb. (**D**) Representative Western blot analyses for HO induction and subsequent degradation (tagged with Flag epitope) and the DSB sensing through Rad53 hyperphosphorylation. A Pgk1 Western blot served as a housekeeping, and Ponceau S staining of the membrane as a loading control for all lanes. The leftmost lane in the Ponceau S corresponds to the protein weight marker. (**E**) Representative Southern blot for the *MAT* switching assay in late mitosis. Alongside the *MAT* probe, a second probe against the *ACT1* gene (1.1 kb fragment) was included for normalization. The *MAT* probe also recognizes an allele-independent *MAT* distal fragment (2.2 kb). (**F**) Quantification of relative band intensities in the *MAT* switching Southern blots (mean ± SEM, n = 3). Individual values were normalized to the *ACT1* signals. Then, every lane was normalized to *MAT*a at the arrest. Tel: telophase. +βE: β-estradiol addition.

The online version of this article includes the following source data and figure supplement(s) for figure 3:

**Source data 1.** Original blots.

**Source data 2.** Uncropped blots.

**Source data 3.** Values for plots.

*Figure 3 continued on next page*

**Figure supplement 1.** The principle of the qPCR resection assay.

**Figure supplement 2.** Features of homologous recombination (HR) to repair the *HOcs* double-strand break (DSB) in late mitosis.

cells, in which HR-driven repair of the HO DSB appeared more robust than other types of DSBs (*Lazzaro et al., 2008*; *Moreau et al., 2001*).

Overall, we concluded that HR is the chosen repair pathway in late mitosis. This is in agreement with predictions based on CDK activity and also fits well with the genetic and cytological data we have obtained before (*Ayra-Plasencia and Machín, 2019*; *Machín and Ayra-Plasencia, 2020*; *Medina-Suárez et al., 2024*). By contrast, this emphasizes the paradoxes of having HR in the context of mostly segregated sister chromatids and without the cohesin complex holding them together as expected. We have now explored the second paradox further as we have just shown that cohesin is reconstituted after DSBs in telophase, and DI-cohesion appears important for efficient DSB repair by HR (*Hou et al., 2022*; *Phipps and Dubrana, 2022*).

## The MAT switching in late mitosis does not depend on cohesin

In cells arrested in *cdc15-2*, the Smc1–Smc3 pool mostly resides at centromeres where it brings together adjacent regions in both chromosome arms (*Garcia-Luis et al., 2022*). Of note, this configuration could favor the physical interaction in cis of the *MAT* and the *HML* loci (*Piazza et al., 2021*), which may in turn promote a rapid and efficient HR-driven *MAT* switching. Thus, we tested whether *MAT* switching was dependent on this residual centromeric cohesin. To this aim, we tagged the Smc3 subunit with the auxin-mediated degron system (aid*). To check that the system was working correctly, a serial dilution spot assay was carried out (*Figure 4A*). The strain expressing Smc3-aid* was plated on YPDA and YPDA plus 8 mM of the auxin indole-acetic acid (IAA). Alongside, we also plated a derivative strain where we had removed the preceptive F-box protein OsTIR1, which forms the functional ubiquitin ligase responsible for targeting the aid* for degradation upon IAA addition (*Morawska and Ulrich, 2013*). Since cohesin is an essential complex for vegetative growth, the lack of growth in the *SMC3-AID* OsTIR1* strain strongly points out that Smc3-aid* is degraded efficiently.

Next, we used this strain to degrade Smc3 after cells have been arrested in late mitosis and before the HO DSB. After IAA addition, there was a clear decline in Smc3-aid* levels, including the chromatin-bound acSmc3-aid* pool (*Figure 4—figure supplement 1A*). After Smc3 depletion, a 1-hr pulse of HO expression was applied, which allowed us to follow both DSB generation and its subsequent repair without residual cohesin. More than 90% of the Smc3-aid* had disappeared by the time the HO promoter was shut down (*Figure 4—figure supplement 1A*). The addition of IAA drastically delayed gene conversion from *MAT*a to *MAT*α (*Figure 4—figure supplement 1B,C*), although the cut efficiency was lower than in the wild type in the absence of auxin (~50% *MAT*a remained uncut in the *SMC3:aid* OsTIR1* strain). Also, the detection of DNA damage through Rad53 phosphorylation seems fainter and slightly delayed (*Figure 4—figure supplement 1*). To unequivocally determine whether *MAT* switching kinetics were affected by Smc3 depletion, and not by an off-target effect of IAA, we compared strains that could either degrade or retain Smc3 in the presence of IAA. Four isogenic strains were analyzed, comprising all possible combinations of Smc3-aid* and OsTir1, as confirmed by Western blot (*Figure 4B*; Tel samples). The addition of IAA-induced Smc3-aid* degradation only in the strain co-expressing OsTIR1 (*Figure 4B*; +IAA samples). In the wild-type strain (no tagged Smc3 and no OsTIR1), the cutting efficiency of *HOcs* and the yield of gene conversion to *MAT*α were both reduced in the presence of IAA, confirming that IAA itself negatively impacts *MAT* switching (*Figure 4C*, compared two leftmost lanes with *Figure 3E*). However, when the yield of the *MAT* conversion after the recovery time was normalized to the cut *HOcs* when HO was repressed, no significant differences were observed among the four strains (*Figure 4C, D*). We therefore conclude that the core of the cohesin complex is not required for the HR-driven repair of the HO DSB in telophase. Of note, it was previously reported that cohesin was not required for the *MAT* switching in

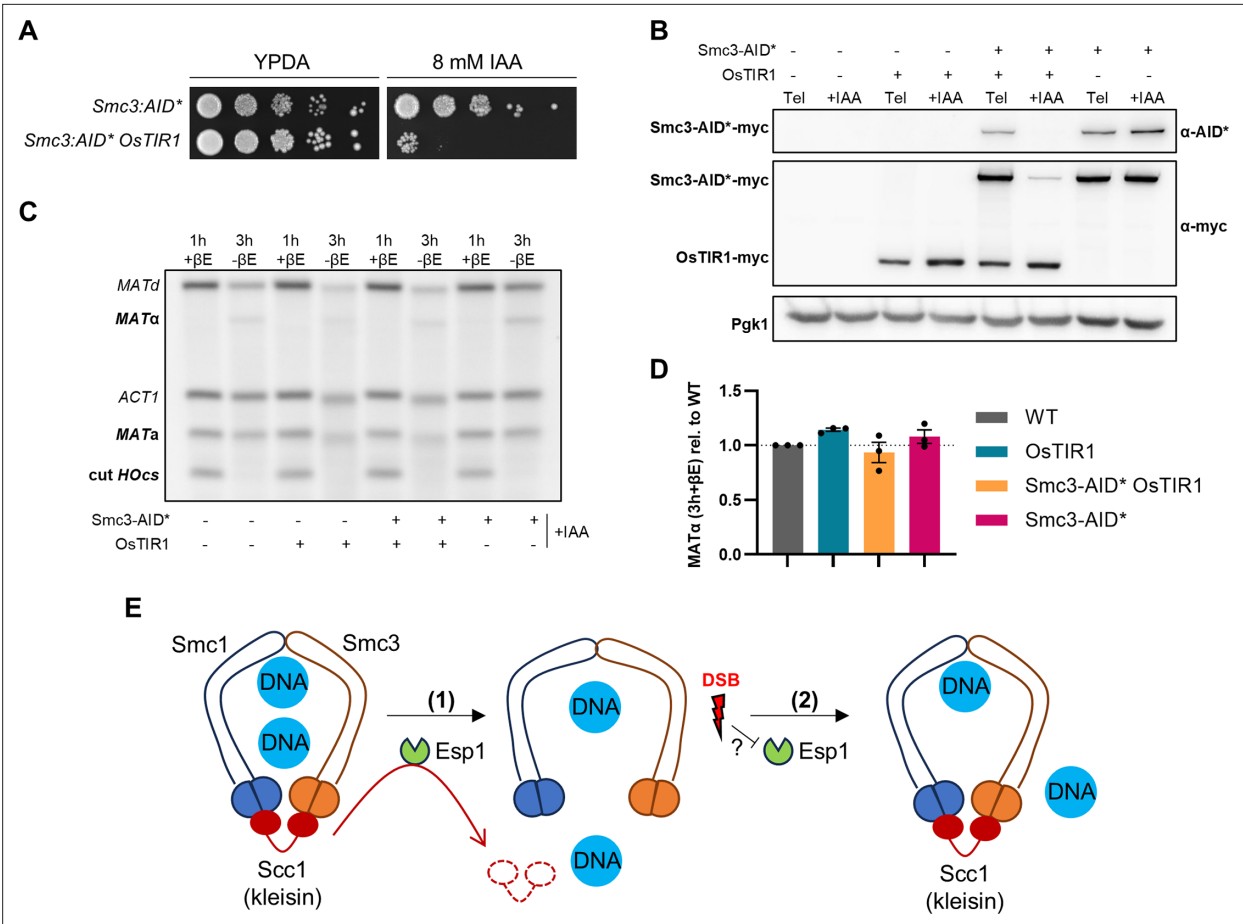

**Figure 4.** Cohesin removal does not alter homologous recombination (HR) at the *MAT* locus in late mitosis. (**A**) Serial dilution spot assay for the *smc3:aid\** strains. The fitness effect of Smc3-aid\* degradation was tested in the *smc3:aid\* OsTIR1* strain in the presence of 8 mM IAA. The same strain without the ubiquitin-ligase *OsTIR1* was also used as the control. (**B–D**) Cells of four strains with all possible combinations of the *smc3:aid\** and *OsTIR1* pairs were treated as in *Figure 1A* but including a 1-hr IAA step between the *cdc15-2* arrest and the induction of HO. Then, HO was induced for 1 hr with β-estradiol (1h+βE sample), after which it was washed off and the cells were allowed to recover from the double-strand break (DSB) for 2 hr (3h-βE sample). IAA and the Tel arrest were maintained throughout the experiment. (**B**) Representative Western blot of the four strains at the time of the arrest (Tel) and 1 hr after IAA addition (+IAA). Note the confirmation of the different genotypic combinations, as well as the Smc3-aid\* decline after IAA addition. (**C**) Representative Southern blot for the *MAT* switching assay of the four strains in the presence of 8 mM IAA. Note that HO cutting is less efficient than without IAA (see *Figure 3E*, *Figure 4—figure supplement 1B, C*) but gene conversion is still possible after degrading Smc3-aid\* in late mitosis. (**D**) Quantification of relative *MAT* switching (mean ± SEM, *n* = 3). Gene conversion to *MATα* in the 3h-βE sample was normalized to the cut *HOcs* in the 1h+βE sample. Then, gene conversion yield was normalized to the wild-type (no tagged Smc3, no OsTir1) strain. Tel: telophase. +βE: β-estradiol addition. −βE: β-estradiol removal. +IAA: indole-acetic acid. (**E**) Model of how DSBs reconstitute cohesin in late mitosis. The cohesin complex (Smc1–Smc3–Scc1) entraps sister chromatids before anaphase. (1) At anaphase onset, Esp1 (separase) is activated to cleave Scc1 and release sister chromatids for segregation. (2) If DSBs occur before cytokinesis, Scc1 returns and reconstitutes the cohesin complex. At least a fraction of the complex binds to chromatin. The reappearance of Scc1 likely occurs through the inhibition of Esp1 and/or post-translational protection of Scc1 by the DNA damage checkpoint.

The online version of this article includes the following source data and figure supplement(s) for figure 4:

**Source data 1.** Original blots.

**Source data 2.** Uncropped blots.

**Source data 3.** Values for plots.

**Figure supplement 1.** MAT switching yield of Smc3-aid\* OsTIR1 upon IAA addition.

**Figure supplement 1—source data 1.** Original blots.

**Figure supplement 1—source data 2.** Uncropped blots.

**Figure supplement 1—source data 3.** Values for plots.

**Table 1.** Strains used in this work.

| Strain | Relevant genotype* | Origin | Figure |
|---|---|---|---|
| FM2344 | (YPH499) *MATa ura3-52 lys2-801 ade2-101 trp1-Δ63 his3-Δ200 leu2-Δ1 bar1-Δ; cdc15-2:9myc::Hph; HMLα Δhmr::HIS3MX* | Machín lab | (Parental) |
| LSY4319-1C | (W303) *MATα leu2-3,112 trp1-1 can1-100 ura3-1 ade2-1 his3-11,15 RAD5 Δhml Δhmr; leu2-3::lexO4p:HO:FLAG:CYC1t::ACT1p:LexA-ER-B112-T:CYC1p::LEU2MX (pRG646)* | Symington lab | (Parental) |
| FM2450 | LSY4319-1C; *cdc15-2:9myc::Hph* | This work | 3A, B |
| FM2520 | FM2450; *SCC1:3myc::HIS3MX* | This work | 1B, E |
| FM2531 | FM2344; *leu2-3::lexO4p:HO:FLAG:CYC1t::ACT1p:LexA-ER-B112-T:CYC1p::LEU2MX* | This work | 3D–F; 4B–D |
| FM2635 | FM2450; *SMC1:6HA:HIS3MX* | This work | 1C, E |
| FM2662 | FM2531; *Δrad9::KanMX4* | This work | 3-fs2 |
| FM2663 | FM2531; *Δyku70::KanMX4* | This work | 3-fs2 |
| FM2668 | FM2531; *Δmre11::KanMX4* | This work | 3-fs2 |
| FM2672 | FM2531; *SMC3:AID*:9myc::KanMX* | This work | 4A–D |
| FM2674 | FM2450; *SMC3:3HA::HIS3MX* | This work | 1D; 2C; 2-fs2; 2-fs3 |
| FM2680 | FM2672; *ura3-52::ADH1p:OsTIR1:9myc::URA3* | This work | 4A–D; 3-fs3 |
| FM3235 | FM2520; *SMC1:6HA::NatNT2* | This work | 2A, B; 2-fs1 |
| FM3294 | FM2531; *ura3-52::ADH1p:OsTIR1:9myc::URA3* | This work | 4B–D |

*Semicolons separate genetic modifications obtained through sequential transformation steps. Intermediate strains are omitted.

G2/M-arrested cells, despite it being essential for efficient post-replicative DSB repair (*Unal et al., 2004*).

## Conclusions and perspectives

In this work, we have addressed whether HR was molecularly efficient in yeast cells arrested in late mitosis. We have used the well-established *MAT* switching system and found that HR is indeed working in late mitosis, complementing previous results that pointed in this direction both genetically and cytologically (*Ayra-Plasencia and Machín, 2019*). In addition, we have addressed the paradox of having HR in a cell mitotic stage with no sister chromatid cohesion. Our results suggest that residual cohesin, or at least residual chromatin-bound Smc1–Smc3 dimers, are complemented by incorporating de novo Scc1 (*Figure 4E*). This fact suggests that chromosome structure may undergo significant changes after late DSBs. The nature of these changes remains unexplored, and we cannot currently rule out that some kind of DI-cohesion could be established, perhaps globally. Alternatively, telophase chromatin may acquire new intramolecular loops as it occurs in G2/M-arrested cells (*Piazza et al., 2021*). Either of these may favor repair by HR with the intact sister or intramolecularly, although it seems still dispensable for *MAT* switching at this late cell cycle stage. Further research will be needed to confirm these hypotheses, as well as the whereabouts of cohesin and the consequences for late segregation after DSB repair.

## Materials and methods
### Strains and experimental conditions

*Table 1* contains the *S. cerevisiae* strains used in this work. Genetic backgrounds are either W303 or YPH499. Strain construction was performed by lithium acetate-based transformation on pre-made frozen competent cells (*Knop et al., 1999*).

Strains were cultured overnight in the YPD medium (10 g·l⁻¹ yeast extract, 20 g·l⁻¹ peptone, and 20 g·l⁻¹ glucose) with moderate shaking (180 rpm). The day after, 10–100 µl of grown cells were diluted

into a flask containing an appropriate YPD volume and grown overnight at 25°C again. Finally, the exponentially growing culture was adjusted to $OD_{600}$ = 0.5 to start the experiment.

In general, cells were first arrested in late mitosis by incubating the culture at 34°C for 3 hr (all strains carry the *cdc15-2* thermosensitive allele). Then, the culture was split in two, one subculture remained untreated (mock), and DSBs were generated in the second one. When a G2/M arrest was required, 15 µg·ml⁻¹ nocodazole (Nz; Sigma-Aldrich, M1404) was added to the asynchronous culture, which was then incubated at 25°C for 3 hr (with a 7.5 µg·ml⁻¹ Nz shot at 2 hr). To conditionally degrade the Smc3-aid* variant, 8 mM of the auxin 3-indol-acetic acid (IAA; Sigma-Aldrich, I2886) was added to the arrested cells 1 hr prior to generating DSBs.

DSB generation was accomplished by incubating the culture for 1–3 hr with either 10 µg·ml⁻¹ phleomycin (random DSBs) or 2 µM of β-estradiol (HO-driven DSBs). For the *HOcs* DSB, the corresponding strain harbors an integrative system based on a β-estradiol inducible promoter (*Ottoz et al., 2014*). These strains were constructed by transforming with the pRG464 plasmid.

## Western blot for protein levels and phosphorylation

Western blotting was performed as reported before (*Ayra-Plasencia and Machín, 2019*). The trichloroacetic acid method was used for protein extraction. Total protein was quantified in a Qubit 4 Fluorometer (Thermo Fisher Scientific, Q33227). Proteins were resolved in 7.5% SDS–PAGE gels and transferred to PVDF membranes (Pall Corporation, PVM020C099). The membrane was stained with Ponceau S solution (PanReac AppliChem, A2935) as a loading reference. The following primary antibodies were used for immunoblotting: mouse monoclonal α-HA (1:1000; Sigma-Aldrich, H9658), mouse monoclonal α-myc (1: 5000; Sigma-Aldrich, M4439), mouse monoclonal α-Pgk1 (1:5000; Thermo Fisher Scientific, 22C5D8), mouse monoclonal α-miniaid (1:500; MBL, M214-3), mouse monoclonal α-Flag (1:5000; Sigma-Aldrich, F3165), mouse monoclonal α-Rad53 (1:1000; Abcam, ab166859), and mouse monoclonal α-acSmc3 (1:5000; a gift from Katsuhiko Shirahige). The secondary antibody was a horseradish peroxidase polyclonal goat anti-mouse (from 1:5000 to 1:10,000, depending on the primary antibody; Promega, W4021). Proteins were detected with the ECL chemiluminescence reagent (GE Healthcare, RPN2232), and visualized in a Vilber-Lourmat Fusion Solo S chamber.

## Co-IP for the reassembly of the cohesin complex

For co-IP assays, approximately 100 $OD_{600}$ units from the corresponding conditions were collected. Cell disruption was performed by resuspending the frozen pellets in 200 µl of Buffer A (50 mM HEPES-Na, 150 mM KCl, 1.5 mM $MgCl_2$, 0.5 mM DTT, 0.5% Triton X-100, 1× EDTA-free protease inhibitor cocktail) together with glass beads. Samples were vortexed vigorously for 5 min, and additional lysis cycles were performed as needed, allowing 1 min incubation on ice between rounds. Lysates were transferred to a clean tube. 400 µl of Buffer A was used to wash the beads and pooled with the lysates. Samples were then centrifuged (12,000 rpm, 5 min, 4°C), and the clarified lysates were transferred to new tubes. An aliquot of 70 µl was saved as whole cell extract for Western blot analysis. The remaining lysate was incubated with pre-equilibrated Pierce Anti-HA Magnetic Beads (Thermo Fisher Scientific; 88836) for 2 hr at 4°C on a rotating wheel. After incubation, beads were placed on a magnetic rack and washed five times with 200 µl of IPP150 buffer (10 mM Tris-Cl pH 7.5, 150 mM NaCl, 0.5% Triton X-100).

For elution, beads were resuspended in 25 µl of 1× SR buffer (8% SDS, 0.5 M Tris-Cl pH 6.8) and incubated at 37°C for 4 min. The supernatant was collected using a magnetic rack and transferred to a new tube containing 8.3 µl of 4× SS buffer (20% sucrose, 0.05% bromophenol blue, 0.1% sodium azide) supplemented with β-mercaptoethanol. Samples were boiled for 2 min at 95°C, centrifuged (max speed, 30 s), and analyzed by SDS–PAGE and Western blotting.

## Chromatin fractionation for the chromatin-bound cohesin

Biochemical fractionation of cells was performed as described before with little modifications (*Cuevas-Bermúdez et al., 2020*). Briefly, cells were collected and resuspended in 400 µl of buffer 1 (20 mM HEPES pH 8, 60 mM KCl, 15 mM NaCl, 10 mM $MgCl_2$, 1 mM $CaCl_2$, 0.8% Triton X-100, 1.25 M sucrose, 0.5 mM spermine). Breakage of cell walls was done by adding ~200 mg of glass beads and vortexing for 4 min. After centrifuging at 500 × *g* for 5 min, supernatant was transferred to a new tube and 40 µl were kept as Input. Samples were centrifuged at 18,000 × *g* for 20 min, the supernatant was kept

as S1, whereas the pellet was resuspended in 200 µl of Buffer 1 (50 µl were saved as P1). Samples were centrifuged again at 18,000 × $g$ for 20 min, the supernatant was kept as S2, and the pellet was resuspended in 200 µl of buffer 2 (20 mM HEPES pH 7.6, 45 mM NaCl, 7.5 mM MgCl$_2$, 20 mM EDTA pH 8, 10% glycerol, 1% NP-40, 2 M urea, 0.5 M sucrose, protease inhibitors) and 50 µl were saved as P2. Finally, samples were centrifuged one last time at the same conditions, keeping the supernatant as S3, and the pellet was resuspended in 50 µl of Laemmli buffer, constituting the chromatin-enriched fraction (P3).

## ChIP–qPCR for the cohesin loading onto the HO DSB

For ChIP–qPCR, a total of 100 OD$_{600}$ units of the treated cultures were collected and crosslinked with 1.4% formaldehyde for 15 min. Crosslinking was quenched with glycine (final concentration 125 mM) for 7 min. Cells were harvested by centrifugation (4000 rpm, 1 min), washed with PBS, transferred to screw-cap tubes, and frozen at –80°C.

Frozen pellets were resuspended in 300 µl of IP buffer (150 mM NaCl, 50 mM Tris-HCl pH 7.5, 5 mM EDTA, 1% Triton X-100, 0.05% NP-40) supplemented with 1 mM PMSF and protease inhibitor cocktail (EDTA-free, Roche). Disruption was performed by bead beating with 500 µl of glass beads using a Bertin Precellys homogenizer for nine 20 s cycles at hard setting. Lysates were transferred to new tubes and supplemented with an additional 100 µl of IP buffer containing PMSF and protease inhibitors. Samples were clarified by centrifugation (15,000 rpm, 10 min, 4°C). Pellets were resuspended in 1 ml IP buffer with PMSF and protease inhibitors and sonicated in a Diagenode Bioruptor Plus (30 cycles: 30 s ON 30 s OFF, high power, 4°C). After sonication, samples were centrifuged (15,000 rpm, 10 min), and the supernatant was collected. A 200-µl aliquot was reserved as input. Input DNA was precipitated with 0.3 M sodium acetate and 2.5 volumes of cold ethanol, centrifuged (15,000 rpm, 30 min), washed with 70% ethanol, and air-dried.

For immunoprecipitation, 400 µl of the sonicated chromatin was incubated with 40 µg of anti-HA antibody (Roche, 11666606001) in the Bioruptor at low power for 30 min (30 s ON/30 s OFF cycles). Antibody–chromatin complexes were pelleted (13,000 rpm, 5 min), and the supernatant was incubated with 60 µl of Dynabeads Protein G (Invitrogen), pre-equilibrated in IP buffer, then incubated for 2 hr at 4°C on a rotating wheel, and finally washed five times with IP buffer using a magnetic rack.

Both input and IP samples were resuspended in de-crosslinking buffer (1× TE, 1% SDS, 10 µg·ml$^{-1}$ RNase A, 1 mg·ml$^{-1}$ proteinase K) and incubated overnight at 65°C and purified using High Pure PCR product purification kit (Roche) for the qPCR.

## qPCR for DSB resection

In the resection experiments, qPCR was performed on genomic DNA (gDNA) extracted by the glass beads/phenol method (*Hoffman and Winston, 1987*). Experimental details on the resection assay can be found in *Gnügge et al., 2018*. Briefly, each gDNA sample was divided into two, and one aliquot was digested with StyI-HF (NEB, R3500S). Then, qPCR reactions were mounted using the PowerUp SYBR Green Master Mix (Thermo Scientific, A25741) and run in a Bio-Rad CFX384 Real-Time PCR instrument (10 µl final volume in 384-well block plates).

Resection was calculated as a normalized fraction ($f_{resected}$) to that of the HOcs effectively cut by HO ($f$). These two parameters were calculated by their corresponding formulas (*Gnügge et al., 2018*).

## Southern blot for the MAT switching assay

In this case, gDNA was extracted by digesting with 50 U lyticase (Sigma-Aldrich, L4025) a cell pellet previously resuspended in 200 µl of digestion buffer (1% SDS, 100 mM NaCl, 50 mM Tris-HCl, and 10 mM EDTA). Then, the gDNA was phase separated by phenol:chloroform (PanReac AppliChem, A0944), precipitated with ethanol, resuspended in TE 1X with 10 µg·ml$^{-1}$ RNase A, precipitated again, and resuspended in TE 1X. Then, the gDNA was digested with StyI, the restriction fragments separated on a 1.2% low EEOO LS Agarose gel, and finally Southern blotted. Southern blot was carried out by a saline downwards transference onto a positively charged nylon membrane (Hybond-N+, Amersham-GE; RPN303B) (*García-Luis and Machín, 2014*). DNA probes against *ACT1* and *MAT* loci were made using Fluorescein-12-dUTP Solution (ThermoFisher; R0101) and the Expand High Fidelity PCR System (Roche; 11732641001). Hybridization with fluorescein-labeled probes was performed overnight at 68°C. The following day, the membrane was incubated with an anti-fluorescein antibody coupled to

alkaline phosphatase (Roche; 11426338910), and the chemiluminescent signal detected using CDP-star (Amersham; RPN3682). Blots were visualized in a Vilber-Lourmat Fusion Solo S chamber.

For band quantification, each individual band was normalized to the *ACT1* signal in the lane. Then, a second normalization to the *MAT*a band at the arrest was performed. To determine the yield of gene conversion to *MAT*α specifically, the *MAT*α band after recovery from the DSB (2–3 hr after β-estradiol removal) was normalized to the amount of the cut *HOcs* band immediately before recovery (1 hr after β-estradiol addition).

## Data representation and statistics

Three types of graphs were used to represent the data: bar charts, marker line graphs, and box plots. In box plots, the center line represents the medians, box limits represent the 25th and 75th percentiles, the whiskers extend to the 5th and 95th percentiles, and the dots represent outliers. Error bars in all bar and line charts represent the SEM of three independent biological replicates. Individual values are also represented as dots in the bar charts. GraphPad Prism 9 was used for generating the charts and for statistical analysis. Differences between experimental data points were generally estimated using the Mann–Whitney *U* test for box plots and one-way ANOVA with Tukey post hoc for bar charts.

## Acknowledgements

We are grateful to Katsuhiko Shirahige for the anti-acSmc3 antibody. We would like to thank other members from Machín's and Symington's labs for enriching and fruitful discussion. This work was supported by the Spanish Ministry of Science, Innovation and Universities (MICIU/AEI/10.13039/501100011033; research grants BFU2017-83954-R and PID2021-123716OB-I00 to FM; co-funded by the EU-ERDF 'A way of making Europe') and the National Institutes of Health (research grant NIH R35 GM126997 to LS). The Agencia Canaria de Investigación, Innovación y Sociedad de la Información (ACIISI) supported SM-S through a predoctoral fellowship (TESIS2020010028; co-funded by the ESF+), and the Servicio Público de Empleo Estatal (SEPE) supported EH-C through an INVES-TIGO program contract (co-funded by the ESF+).

## Additional information

### Funding

| Funder | Grant reference number | Author |
|---|---|---|
| Ministerio de Ciencia, Innovación y Universidades | BFU2017-83954-R | Félix Machín |
| Ministerio de Ciencia, Innovación y Universidades | PID2021-123716OB-I00 | Félix Machín |
| National Institutes of Health | NIH R35 GM126997 | Lorraine S Symington |

The funders had no role in study design, data collection, and interpretation, or the decision to submit the work for publication.

### Author contributions

Jessel Ayra Plasencia, Data curation, Formal analysis, Investigation, Visualization, Methodology, Writing – original draft, Writing – review and editing; Sara Medina-Suárez, Esperanza Hernández-Carralero, Jonay García-Luis, Formal analysis, Investigation, Visualization, Methodology, Writing – review and editing; Lorraine S Symington, Supervision, Funding acquisition, Project administration, Writing – review and editing; Félix Machín, Conceptualization, Resources, Data curation, Formal analysis, Supervision, Funding acquisition, Visualization, Writing – original draft, Project administration, Writing – review and editing

### Author ORCIDs

Jessel Ayra Plasencia https://orcid.org/0000-0003-1052-4214
Sara Medina-Suárez https://orcid.org/0000-0002-3612-764X

Esperanza Hernández-Carralero [ORCID] https://orcid.org/0000-0002-0653-3583
Jonay García-Luis [ORCID] http://orcid.org/0000-0002-2799-7491
Lorraine S Symington [ORCID] https://orcid.org/0000-0002-1519-4800
Félix Machín [ORCID] https://orcid.org/0000-0003-4559-7798

Reviewer #1 (Public review): https://doi.org/10.7554/eLife.92706.3.sa1
Reviewer #2 (Public review): https://doi.org/10.7554/eLife.92706.3.sa2
Author response https://doi.org/10.7554/eLife.92706.3.sa3

## Additional files

### Supplementary files
MDAR checklist

### Data availability
All data generated or analyzed during this study are included in the manuscript and supporting files; source data files have been provided for all figures.

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

# Appendix 1

## Appendix 1—key resources table

| Reagent type (species) or resource | Designation | Source or reference | Identifiers | Additional information |
|---|---|---|---|---|
| Strain, strain background (*Saccharomyces cerevisiae*, W303) | FM2450 | This work | | See *Table 1* |
| Strain, strain background (*Saccharomyces cerevisiae*, W303) | FM2520 | This work | | See *Table 1* |
| Strain, strain background (*Saccharomyces cerevisiae*, YPH499) | FM2531 | This work | | See *Table 1* |
| Strain, strain background (*Saccharomyces cerevisiae*, W303) | FM2635 | This work | | See *Table 1* |
| Strain, strain background (*Saccharomyces cerevisiae*, YPH499) | FM2662 | This work | | See *Table 1* |
| Strain, strain background (*Saccharomyces cerevisiae*, YPH499) | FM2663 | This work | | See *Table 1* |
| Strain, strain background (*Saccharomyces cerevisiae*, YPH499) | FM2668 | This work | | See *Table 1* |
| Strain, strain background (*Saccharomyces cerevisiae*, YPH499) | FM2672 | This work | | See *Table 1* |
| Strain, strain background (*Saccharomyces cerevisiae*, W303) | FM2674 | This work | | See *Table 1* |
| Strain, strain background (*Saccharomyces cerevisiae*, YPH499) | FM2680 | This work | | See *Table 1* |
| Strain, strain background (*Saccharomyces cerevisiae*, W303) | FM3235 | This work | | See *Table 1* |
| Strain, strain background (*Saccharomyces cerevisiae*, YPH499) | FM3294 | This work | | See *Table 1* |
| Chemical compound, drug | Nocodazole | Sigma-Aldrich | M1404 | |
| Chemical compound, drug | Auxin 3-indol-acetic acid (IAA) | Sigma-Aldrich | I2886 | |
| Chemical compound, drug | Phleomycin | Sigma-Aldrich | P9564 | |
| Chemical compound, drug | β-Estradiol | Sigma-Aldrich | E8875 | |
| Chemical compound, drug | Trichloroacetic acid | Sigma-Aldrich | T4885 | |
| Chemical compound, drug | Ponceau S solution | PanReac AppliChem | A2935 | |

*Appendix 1 Continued on next page*

*Appendix 1 Continued*

| Reagent type (species) or resource | Designation | Source or reference | Identifiers | Additional information |
|---|---|---|---|---|
| Chemical compound, drug | ECL chemiluminescence reagent | GE Healthcare | RPN2232 | |
| Chemical compound, drug | EDTA-free protease inhibitor cocktail | Roche | 5892791001 | |
| Chemical compound, drug | PMSF | Roche | 10837091001 | |
| Chemical compound, drug | RNase A | Roche | 10109169001 | |
| Chemical compound, drug | Proteinase K | Roche | 3115879001 | |
| Chemical compound, drug | β-Mercaptoethanol | Sigma-Aldrich | M3148 | |
| Chemical compound, drug | Laemmli buffer | Bio-Rad | 1610747 | |
| Chemical compound, drug | Formaldehyde | Sigma-Aldrich | 47608 | |
| Chemical compound, drug | Glycine | Promega | H5073 | |
| Chemical compound, drug | NP-40 | Sigma-Aldrich | I8896 | |
| Chemical compound, drug | Triton X-100 | Promega | H5141 | |
| Chemical compound, drug | StyI-HF | NEB | R3500S | |
| Chemical compound, drug | Lyticase | Sigma-Aldrich | L4025 | |
| Chemical compound, drug | Phenol:chloroform | PanReac AppliChem | A0944 | |
| Chemical compound, drug | Fluorescein-12-dUTP Solution | Thermo Scientific | R0101 | |
| Chemical compound, drug | CDP-star | Amersham | RPN3682 | |
| Chemical compound, drug | Sucrose | PanReac AppliChem | 571621.1611 | |
| Commercial assay, kit | High Pure PCR product purification kit | Roche | 11732676001 | |
| Commercial assay, kit | Expand High Fidelity PCR System | Roche | 11732641001 | |
| Commercial assay, kit | PowerUp SYBR Green Master Mix | Thermo Scientific | A25741 | |
| Other | Low EEOO LS Agarose | PanReac AppliChem | 374114.1209 | |
| Other | Positively charged nylon membrane (Hybond-N+) | Amersham GE | RPN303B | |
| Other | PVFD membranes | Pall Corporation | PVM020C099 | |
| Other | Pierce Anti-HA Magnetic Beads | Thermo Fisher Scientific | 88836 | |
| Other | Dynabeads Protein G | Invitrogen | 10003D | |
| Other | Qubit 4 Fluorometer | Thermo Fisher Scientific | Q33227 | |

*Appendix 1 Continued on next page*

*Appendix 1 Continued*

| Reagent type (species) or resource | Designation | Source or reference | Identifiers | Additional information |
|---|---|---|---|---|
| Other | Real-Time PCR instrument | Bio-Rad | CFX384 | |
| Antibody | Mouse monoclonal anti-HA | Sigma-Aldrich | H9658; RRID:AB_260092 | 1:1000 |
| Antibody | Mouse monoclonal anti-myc | Sigma-Aldrich | M4439; RRID:AB_439694 | 1:5000 |
| Antibody | Mouse monoclonal anti-Pgk1 | Thermo Fisher Scientific | 22C5D8; RRID:AB_2532235 | 1:5000 |
| Antibody | Mouse monoclonal anti-miniaid | MBL | M214-3; RRID:AB_2890014 | 1:500 |
| Antibody | Mouse monoclonal anti-Rad53 | Abcam | ab166859; RRID:AB_2801547 | 1:1000 |
| Antibody | Horseradish peroxidase polyclonal goat anti-mouse | Promega | W4021; RRID:AB_430834 | 1:5000 to 1:10,000 |
| Antibody | anti-HA antibody | Roche | 11666606001; RRID:AB_514506 | For ChIP–qPCR |
| Antibody | Anti-fluorescein antibody coupled to alkaline phosphatase | Roche | 11426338910; RRID:AB_514504 | |
| Software, algorithm | BioProfile Bio1D | Vilber-Lourmat | v15.07 | Vilber-Lourmat Fusion Solo S chamber |
| Software, algorithm | Prism | GraphPad | v9; RRID:SCR_002798 | |
| Recombinant DNA reagent (plasmid) | pNHK53 | Kanemaki lab | | ADH1p-OsTIR-9Myc (URA) |
| Recombinant DNA reagent (plasmid) | pRG464 | Symington lab | | LexA-TF-PlexOp:HO::LEU2 |
| Sequence-based reagent | SMC1-S2 | This work | GTCGAAGATCATAACTTTGGACTTGAGCAATTACGCAGAACGTACGCTGCAGGTCGAC | PCR-primer for C-terminal tagging |
| Sequence-based reagent | SMC1-S3 | This work | TTATTTGACGGGTTATAGCAGAGGTTGGTTTCATAGATTAATCGATGAATTCGAGCTCG | PCR-primer for C-terminal tagging |
| Sequence-based reagent | SMC3-S2 | This work | AATCGGATTCATTAGAGGTAGCAATAAATTCGCTGAAGTCCGTACGCTGCAGGTCGAC | PCR-primer for C-terminal tagging |
| Sequence-based reagent | SMC3-S3 | This work | ACTGATATTTTTATATACAAATCGTTTCAAATATCTCTTAATCGATGAATTCGAGCTCG | PCR-primer for C-terminal tagging |
| Sequence-based reagent | SCC1-S2 | This work | ATCAGCTTATTGGGTCCACCAAGAAATCCCCTCGGCGTAACTAGGTTTTAATCGATGAATTCGAGCTCG | PCR-primer for C-terminal tagging |
| Sequence-based reagent | SCC1-S3 | This work | ATATTAAAATAGACGCCAAACCTGCACTATTTGAAAGGTTTATCAATGCTCGTACGCTGCAGGTCGAC | PCR-primer for C-terminal tagging |
| Sequence-based reagent | cdc15-F(−132) | This work | TCTTTCCGCTTTTCTTGCTG | PCR-primer for allele transfer |
| Sequence-based reagent | cdc15-R(+3023) | This work | TGCGTTTTCAGTATTGGAAGG | PCR-primer for allele transfer |
| Sequence-based reagent | Rad9-F(−326) | This work | GCAGCTCCCCATCAAAATAA | PCR-primer for allele transfer |
| Sequence-based reagent | Rad9-R(+4158) | This work | TCATTACAAGATGCAAGCCTAAA | PCR-primer for allele transfer |
| Sequence-based reagent | yku70-F(−361) | This work | TCCGTTTTGACAACAGGTCACTTCT | PCR-primer for allele transfer |
| Sequence-based reagent | Yku70+300 | This work | CCACAAAGTAATTGTCAGGAAGTGGAAACCCTTG | PCR-primer for allele transfer |
| Sequence-based reagent | Mre11-F(−282) | This work | TCATTGTAGGCATGCACGTT | PCR-primer for allele transfer |
| Sequence-based reagent | Mre11-R(+2258) | This work | ACAAAAGAGCAAAGGCTGGA | PCR-primer for allele transfer |

*Appendix 1 Continued on next page*

*Appendix 1 Continued*

| Reagent type (species) or resource | Designation | Source or reference | Identifiers | Additional information |
|---|---|---|---|---|
| Sequence-based reagent | HO-150-F | This work | TCGTGGCGGAGGTTGTTTAT | PCR-primer for ChIP–qPCR |
| Sequence-based reagent | HO-150-R | This work | ACAAAAGAGGCAAGTAGATAAGGGT | PCR-primer for ChIP–qPCR |
| Sequence-based reagent | HO-500-F | This work | GGACGGATGACAAATGCACC | PCR-primer for ChIP–qPCR |
| Sequence-based reagent | HO-500-R | This work | TGAAGCCGAAGGTAACTAGCA | PCR-primer for ChIP–qPCR |
| Sequence-based reagent | HO-1.5kb-F | This work | ACATTTTCAATCAAGCTGCGGA | PCR-primer for ChIP–qPCR |
| Sequence-based reagent | HO-1.5kb-R | This work | AATGTCCAAAATTGGTGAAGCA | PCR-primer for ChIP–qPCR |
| Sequence-based reagent | HO-3kb-F | This work | GCAAGTGCCCATGCTAACTC | PCR-primer for ChIP–qPCR |
| Sequence-based reagent | HO-3kb-R | This work | CCTACCGCACCTTCTAAGCA | PCR-primer for ChIP–qPCR |
| Sequence-based reagent | HO-10kb-F | This work | TCCTTCGCAACTTTCCTCCC | PCR-primer for ChIP–qPCR |
| Sequence-based reagent | HO-10kb-R | This work | GTGTGACCATGGACGAGGAG | PCR-primer for ChIP–qPCR |

