## [Editor Report · eLife Assessment]

This study provides **convincing** evidence that homologous recombination can occur in telophase-arrested cells, independently of cohesin subunits Smc 1-3. These findings are **valuable** as they point to investigate the role of cohesins re-association with chromatin in the allelic inter-sister repair by homologous recombination.

---

## [Referee Report · Reviewer #1 (Public review)]

Summary

The cohesin complex is essential for maintaining sister chromatid cohesion from S phase until anaphase. Beyond this canonical role, it is also recruited to double-strand breaks (DSBs), supporting both local and global post-replicative cohesion, a phenomenon first reported in 2004. In a previous study, Ayra-Plasencia et al. demonstrated that in telophase, DSBs can be repaired by homologous recombination (HR) through re-coalescence of sister chromatids (Ayra-Plasencia & Machín, 2019). In the present work, the authors provide further insights into DSB repair in late mitosis, showing that:

Scc1 is reloaded and reconstituted on chromatin together with Smc1.

HR occurs with high efficiency.

HR-driven MAT switching can occur in an Smc3-independent manner.

Strengths

The authors take full advantage of the yeast model system, employing the HO endonuclease to generate a single, site-specific DSB at the MAT locus on chromosome III. Combined with careful cell synchronization, this setup allows them to monitor HR-mediated repair events specifically in G2/M and late mitosis. Their demonstration that full-length Scc1 can be recovered upon DSB induction is compelling. Most importantly, the finding that efficient HR can take place during M phase is significant, as HR has long been thought to be largely inhibited at this stage of the cell cycle.

Weaknesses

While the authors provide evidence for Scc1 recovery and efficient HR in late mitosis, some critical points need to be clarified to improve the impact and interpretability of the study.

---

## [Referee Report · Reviewer #2 (Public review)]

Cohesin drive inter-sister repair of DNA breaks by homologous recombination (HR) in G2/M. Cohesion is lost at the metaphase to anaphase transition upon digestion of the Scc1 subunit of cohesin by Esp1, raising the question as to whether and how break repair by HR could occur in late mitosis (late-M).

Here the author investigate the behavior of cohesin in cells arrested in telophase and experiencing a DNA break at the mating-type locus on chr. III (a specialized recombination process required for mating-type switching) or upon random DNA break formation with the drug phleomycin.

The revised version of the manuscript now convincingly establishes three facts:

- The cohesin subunit Scc1 can re-associate with chromatin and the other Smc1-3 subunits upon formation of an unrepairable DSB at MAT in telophase.

- HR can occur in telophase-arrested cells

- Cohesin (an a fortiori cohesin that reassociated with chromatin) plays no role in non-allelic HR in telophase in the specific context of MAT switching.

Unfortunately, the role of cohesin re-association with chromatin for the allelic inter-sister repair by HR is not addressed. In the absence of such evidence, the main claims of the paper making up the title (cohesin re-association and HR repair) appear disconnected. Even if the very last sentence of the abstract corrects the false sense from the title and the rest of the abstract that cohesin reconstitution has somehow something to do with efficient HR in late mitosis, I think a general rewriting of the abstract and a different title would better lift any ambiguity about the conclusions of the paper.

---

## [Author Response]

The following is the authors’ response to the original reviews

We would like to thank the reviewers for taking the time to thoroughly revise our work. We have considered their suggestions carefully and tried our best to respond to them point by point. Based on their recommendations, two major issues came forward: (1) the strength of our claims about the involvement of cohesin in HR-driven repair in late mitosis; and (2) the underlying mechanism that reconstitutes cohesin in late mitosis after DNA damage. In this revision, we focused on the former and left the latter out (yet it is discussed). We considered that the question of how cohesin returns in late mitosis after DNA damage is important and worthy of further research, but it is beyond the scope of this study (as it is the putative role of condensin). Thus, we have focused on buttressing our main claims, as otherwise pointed out by the reviewers. What have we done to strengthen the role of cohesin in late mitotic DSB repair?

(1) We have biologically replicated and quantified the reappearance of Scc1 after DSB generation (new Figure 1e). We have also quantified changes for the other core subunits (new Figure 1c-e).

(2) We now show that the newly synthetized Scc1 serves to assemble back the cohesin complex (new Figure 2a and S1).

(3) We have performed chromatin fractionation and show that cohesin binding to chromatin increases after the HO-induced DSB (new Figure 2b and S2).

(4) We have performed ChIP assays and show that, despite the increase in the chromatin-bound fraction, the *HOcs* DSB does not recruit new cohesin to the locus (new Figure 2c and S3).

(5) A key assertion in the preprint version was that depleting cohesin using the auxin degron system impairs HR-driven MAT switching. This claim was based on a direct comparison of cultures treated or not with auxin (-/+ IAA). However, during the revision process, we realized that auxin treatment itself could interfere with MAT switching. Firstly, we noticed a diminished *HOcs* cutting efficiency by HO in +IAA cultures (Figure S6). Secondly, the apparently dramatic delay in gene conversion to *MAT*α could actually be related to other undesirable effects of IAA downstream in the repair process. Thus, we decided to repeat this experiment with strains that differ in their response to auxin, so that we could compare all strains in the presence of auxin. We compared four isogenic strains: *SMC3*; *SMC3-aid**; *SMC3 + OsTIR1*; and *SMC3-aid* + OsTIR1*. As a result, we can now show that cohesin depletion does not affect MAT switching (see new Figure 4b-d).

(6) We recently reported a negative chemical interaction between auxin and phleomycin. Auxin appears to diminish the ability of phleomycin to generate DSBs (Comm Biol 2025, doi: 10.1038/s42003-025-08416-x; see Figures S14 and S15 in that paper). While the underlying nature of this interaction is unknown to us (we are working on it), this leads us to omit the coalescence assay included in the preprint version (old Figure 4c), as the diminished coalescence upon IAA addition is actually due to this effect rather than cohesin depletion. This is also in agreement with the new data we include in the revised version, in which we observed only minor changes in cohesin reconstitution and chromatin binding after phleomycin (Figure 2a,b; S1 and S2).

(7) In addition to addressing these reviewers’ requests, we have better characterized the *MAT* switching in late mitosis by incorporating the kinetics of *rad9*Δ (deficient in the DNA damage checkpoint), *yku70*Δ (deficient in non-homologous end joining) and *mre11*Δ (deficient in DSB end tethering). The effect of *rad52*Δ (deficient in HR) has been described elsewhere (our iScience 2024, 10.1016/j.isci.2024.110250).

As a result of these new experiments, new figure panels have been added in the main figures and as supplementary figures. To make room for the these panels in the main figures and keep the short report format, the following changes have been made: (i) old figures and new panels have been combined into four main figures, (ii) some panels from the old figures have been moved to supplementary figures, and (iii) some panels have been reordered for the sake of simplicity and fluidity in the main text.

**Public Reviews:**

**Reviewer #1 (Public Review):**
Summary:The cohesin complex maintains sister chromatid cohesion from S phase to anaphase. Beyond that, DSBs trigger cohesin recruitment and post-replication cohesion at both damage sites and globally, which was originally reported in 2004. In their recent study, Ayra-Plasencia et al reported in telophase, DSBs are repaired via HR with re-coalesced sister chromatids (Ayra-Plasencia & Machín, 2019). In this study, they show that HR occurs in a Smc3-dependent way in late mitosis.Strengths:The authors take great advantage of the yeast system, they check the DSB processing and repair of a single DSB generated by HO endonuclease, which cuts the MAT locus in chromosome III. In combination with cell synchronization, they detect the HR repair during G2/M or late mitosis. and the cohesin subunit SMC3 is critical for this repair. Beyond that, full-length Scc1 protein can be recovered upon DSBs.Weaknesses:These new results basically support their proposal although with a very limited molecular mechanistic progression, especially compared with their recent work.
**Reviewer #2 (Public Review):**
Summary:The manuscript "Cohesin still drives homologous recombination repair of DNA double-strand breaks in late mitosis" by Ayra-Plasencia et al. investigates regulations of HR repair in conditional cdc15 mutants, which arrests the cell cycle in late anaphase/telophase. Using a non-competitive MAT switching system of *S. cerevisiae*, they show that a DSB in telophase-arrested cells elicits a delayed DNA damage checkpoint response and resection. Using a degron allele of SMC3 they show that MATa-to-alpha switching requires cohesin in this context. The presence of a DSB in telophase-arrested cells leads to an increase in the kleisin subunit Scc1 and a partial rejoining of sister chromatids after they have separated in a subset of cells.Strengths:The experiments presented are well-controlled. The induction systems are clean and well thought-out.Weaknesses:The manuscript is very preliminary, and I have reservations about its physiological relevance. I also have reservations regarding the usage of MAT to make the point that inter-sister repair can occur in late mitosis.

Regarding these two weaknesses:

- Physiological relevance: This is something we already addressed in our previous research work (Nat Commun. 2019; 10(1):2862. doi: 10.1038/s41467-019-10742-8), and which was further discussed in a follow-up theoretical paper (Bioessays. 2020 ;42(7):e2000021. doi: 10.1002/bies.202000021). In summary, this is physiologically relevant because a DSB in anaphase activates a late-mitotic checkpoint so the DSB can be repaired before cytokinesis. The fact that anaphase is quick and only a minor fraction of cells get a DSB in this cell cycle stage in an asynchronous population does not preclude its importance since it is enough a single mis-repaired DSB in hundreds of cells to mutate a population in an health- or evolution-relevant way.

- MAT system in late mitosis: It was not our intention to use the MAT switching assay to state that inter-sister repair can occur in late-M. The purpose was to address whether HR was fully functional in this non-G2/M non-G1 stage. Having said that, it is very challenging to design a strategy based on sequence-specific DSB to tackle the inter-sister repair in late-M. Any endonuclease-generated DSB is going to cut in both sisters. This is something we also deeply discussed in our previous works (Nat Commun & Bioessays).

**Recommendations for the authors:**

**Reviewer #1 (Recommendations For The Authors):**
Major points:(1) Smc3 degradation affects Rad53 activation upon DSBs, and this may directly lead to HR repair deficiency. Smc3 also could be phosphorylated by ATM and functions in DNA damage checkpoint activation, these alternative possibilities should also be tested before addressing the bona fide role of Smc3 in this context.

Our previous data already suggested that Rad53 hyperphosphorylation still occurs after Smc3 degradation (Figure S6). Regardless, the question of whether the DNA damage checkpoint (DDC) may play a distinct role in the MAT switching has been addressed in this revision by comparing *RAD9* versus *rad9*Δ. Rad9 is a mediator in the DDC required for the activation of Rad53. We have seen that MAT switching in *rad9*Δ is as efficient as in *RAD9* (new Figure S5d-f).

On the other hand, our new results, in which we have compared four different strains with all auxin system combinations in the presence of auxin, show that cohesin depletion does not affect MAT switching. Previously, we compared minus versus plus auxin and noticed diminished HO cutting efficiency. Thus, we repeated this experiment with four isogenic strains (*SMC3*; *SMC3-aid**; *SMC3 + OsTIR1*; and *SMC3-aid* + OsTIR1*) that differ in their response to auxin and ability to degrade cohesin, so that we could compare all strains in the presence of auxin. As a result, we can now affirm that cohesin depletion does not affect MAT switching (see new Figure 4b-d). Therefore, HR appears efficient after cohesin depletion.

(2) The requirement of cohesin subunit Smc3 and "coincidently" recovery of Scc1 are not sufficient to claim they act as a cohesin complex in this scenario. CoIP in the chromatin fraction after DSBs to prove the cohesin complex formation is recommended. If they act as a complex, are cohesin loader Scc2/4 required?

We have constructed a *SMC3-HA SCC1-myc* strain. We have purified the chromatin-bound fraction as well as performing the co-IP. We have found Smc1-acSmc3-Scc1 forms a complex after Scc1 returns, and that at least a fraction of this complex binds to the chromatin in our HO model of DSBs in late anaphase (the *cdc15-2* arrest). This is now shown in the new Figures 2a,b and S1,S2.

As for the requirement of Scc2/4, we consider that the mechanisms underlying how Scc1 comes back, how a new cohesin complex is reassembled, and how it can partly bind to the chromatin in late anaphase are beyond the scope of this study and worth pursuing in a follow-up story.

(3) Figure 3b. acetylated SMC3 was prominently detected in the absence of DSBs. During the cohesion cycle, the cohesin was released from chromatin in a separase-dependent manner at the anaphase onset. Released Smc3 was deacetylated by Hos1 subsequently. In principle, the acSMC3 level could be very low in late mitosis.

In that figure (now renumbered as Fig S6), we did detect acetylated Smc3 for the remnant Smc3 still found in late mitosis, however, a direct comparison between the acetylated versus non-acetylated pools was not performed, and would require more sophisticated approaches. Note that blots are distinctly exposed until the band is detected, and that signal intensity is antibody-specific. The presence of an acSmc3 pool in the *cdc15-2* arrest is now further confirmed by the new blots in Figures 2a, S1 and S2b.

On the other hand, previous time course experiments from G1 and G2/M releases point out that Smc3 deacetylation is incomplete in anaphase, with up to 30% of acetylated Smc3 remaining (Beckouët et al, 2010 doi:10.1016/j.molcel.2010.08.008). This is consistent with the presence of acSmc3 in the *cdc15-2* arrest.

(4) Did the author examine the acSMC3 levels returning after DSB, as Scc1's levels? If so, how about the Eco1's protein level? Chromatin fractionation could be conducted to check the chromatin-bound SMC3, acSMC3/Eco1, SCC1, SCC1 phosphorylation, and SMC1. These results will tell us whether cohesin functions in DSB repair in late M in a cohesion state.

As stated above, we have now determined that cohesin depletion does not affect HR-driven MAT switching. As for the other questions, yes, we have performed both an assessment of acSmc3 in the pull down and chromatin fractionation, before and after DSBs (new Figures 2a, S1 and S2b). Interestingly, we have noticed a difference between the HO-generated and the phle-generated DSBs. It appears that the former leads to a better reconstituted Smc1-acSmc3-Scc1 complex and more chromatin-bound cohesin. The overall acSmc3 levels do not appear to significantly change in the whole cell extracts, although there could be further posttranslational modifications in telophase (see the changes in intensity between the two acSmc3 bands in Figure S1).

The role of Eco1 has not been directly addressed but is discussed. The main point here is that Eco1 levels may be low after G2/M (e.g., Lyons and Morgan, 2011), but there is still a significant acSmc3 pool in anaphase as Hof1 does not deacetylate all Smc3 (Beckouët et al., 2010).

(5) Figure 4a, the return of full-length Scc1 is based on a single experiment. What's the mechanism? Inhibition of cleavage or re-expression? How about its mRNA levels?

We have repeated the full-length Scc1 experiment two more times. Now, an expression graph is included as a new Figure 1e. The two other subunits, Smc1 and Smc3, have been assessed as well, with no major changes in abundance (new Figure 1c and d).

We feel that the exact molecular mechanism of how Scc1 returns is beyond the scope of this study, but we discuss that the DDC may either inactivate separase or protect Scc1 against it. Indeed, there is literature that supports both mechanisms (e.g., Heidinger-Pauli et al., 2008 doi:10.1016/j.molcel.2008.06.005; Yam et al., 2020 doi:10.1093/nar/gkaa355).

Minor points:(6) FACS data should be shown for all cell synchronization experiments.

From our previous own works, FACS profiles add little to late-M experiments. To properly confirm late-M, microscopy is a must. FACS cannot differentiate between G2/M (metaphase-like), anaphase, telophase and the ensuing G1 (as *cdc15-2* cells do not immediately split apart after re-entering G1). In all experiments, Tel samples (late-M *cdc15-2* arrest) were characterized by >95% large budded binucleated cells.

(7) Figure 1d, A loading control of Rad53-P in is missing. The "Arrest" samples should be loaded again on the right to confirm the shift of Rad53, but not due to "smiling gels".

It is true that the blot on the right has a right-handed smile; however, it is very clear the presence of the Rad53/Rad53-P partner. Because there is not a full shift from Rad53 to Rad53-P, the concern of misidentifying Rad53-P as a result of a blot smile is unfounded.

(8) Figure 1c, After the HO cut, the resected DNA at the 726 bp site reaches to platform at about 4 hrs, while it still increases at the 5.6 kb site. Thus, it is difficult to conclude that "The time to reach half of the maximum possible resection (t1/2) was ~1 h at 0.7 Kb and ~2.5 h at 5.7 Kb from the DSB, respectively".

We assumed that both loci reach the plateau at 0.8 (which is consistent with other studies), so the t1/2 was calculated when the resected intersected 0.4.

(9) Figure 2b and 2c are wrongly labeled.

We have fixed this (now Fig. 3d and e).

(10) Figure 2d, Double check and make sure the quantitative data reflects the representative result. E.g. in Figure 2b (in fact should be 2c). For instance, in Figure 2b, the MATα signals seem to remain stable from 60' to 180', but they keep increasing in Figure 2d. In Yamaguchi & James E. Haber's paper, the signals and changes of MATa and MATα over time are way stronger compared to this study.

We have double checked this. It is true that the sum of *MAT*α, *MAT*alpha and cut *HOcs* bands throughout the assay does not have the intensity seen for MATa before the HO induction (Tel), but *MAT*alpha and *HOcs* signals cannot be established based on the equimolarity of the reaction as all band signals are probe-specific (the best indication of this can be seen in the signal comparison between *MAT*α and *MAT* distal at Tel). Alternatively, some resected *HOcs* may remain unrepaired.

As for the referred example (now Figure 3e), note that they are double normalized to *ACT1* and *MAT*α (Tel), and the *ACT1* band gets fainter after 60’. This explains the increase in the MATalpha quantification in spite of what is apparently seen in the blot.

(11) Typos and fonts: e.g. lines 111-112; line 76 "his link".

We have fixed this. Thanks.

**Reviewer #2 (Recommendations For The Authors):**
Major concerns:(1) Physiological relevance. The authors show that HR can happen in the anaphase to telophase interval, yet does it outside of an hours-long artificial arrest upon inactivation of Cdc15? It is this reviewer's understanding that the duration of the anaphase to telophase transition is short, in the order of minutes. In fact, break signaling and resection are delayed by ~1 hour (Fig. 1), which suggests that cells avoid dealing with the damage and engaging in HR in the anaphase-telophase interval. Is there any described physiological context or checkpoint that blocks this transition for extended periods, that would make any of the findings in this paper relevant?

This concern about the physiological relevance was addressed in our previous study (Nat Commun. 2019; 10(1):2862. doi: 10.1038/s41467-019-10742-8). In that paper’s Figure 1, we showed that G1 re-entry after a *cdc15-2* release was delayed by several hours when DSBs had been previously generated at the *cdc15-2* arrest. We also showed that such a delay depended on Rad9 (i.e., the DNA damage checkpoint). In addition, synchronized (not arrested) cells transiting through anaphase responded to DSB generation by slowing anaphase transition while partly regressing chromosome segregation (Figure S7 in that paper).

(2) Methodological caveats. It is unclear why the authors chose to study DSB-repair in the context of MATa-to-alpha switching (which uses an ectopic donor on the other chromosome arm) as a model for inter-sister repair. It creates a disconnect in the claims of the paper, which means to study inter-sister repair. Studying the kinetics of DSB repair by cytology following low-dose irradiation or radiomimetic drugs would have been a better option. Phleomycin is used in Fig. 4, but the repair kinetics (e.g. Rad52 foci) is not studied.

The MAT switching assay was used here to address how much HR was functional in late-M compared to G2/M (metaphase-like). Then, it was employed to check how cohesin depletion hampers HR in late-M. Even though this is something we already deeply discussed previously (Nat Commun. 2019; 10(1):2862. doi: 10.1038/s41467-019-10742-8; Bioessays. 2020 ;42(7):e2000021. doi: 10.1002/bies.202000021), it is worth recapitulating the methodological challenges that the study of inter-sister repair has in late-M: (i) endonuclease-based DSBs are going to generate two DSBs, one per sister chromatid; (ii) the use of a homologous chromosome without the cutting site as a template is pointless because a sister of the homolog is always going to co-segregate with the broken chromatid, and the same caveat applies for any other ectopic sequence. In this context, the *MAT*a with the *HML* ectopic intrachromosomal sequence is as valid as any other option, with the advantage that it is a very well-known system.

On the other hand, most of the reviewer’s concerns about the inter-sister repair by cytology and the role of Rad52 was addressed in our previous paper (Nat Commun). Note that our new results about the cohesin role on MAT switching show that this HR-mediated DSB repair does not depend on cohesin (new Figure 4b-d).

(3) Preliminary work. The requirement of cohesin for MAT switching in cdc15 mutants would have warranted several additional experiments. Indeed, Cohesin has been shown to regulate homology search in multiple ways upon DNA damage checkpoint-induced metaphase-arrest (see Piazza et al. Nat Cell Biol 2021 (10.1038/s41556-021-00783-x), not cited in the current manuscript). Consequently, is the effect of cohesin observed in the MAT system specific to telophase or is it true in other cell-cycle phases? What is the mechanism behind this requirement (one may expect it not to depend on the sister since the HML donor is available within the damaged chromatid)? Does cohesin re-accumulate around the DSB site or genome-wide? How does the Esp1 activity decay from anaphase onset? Is cohesin required for the horseshoe folding of chr. III involved in MATa-to-alpha switching? Furthermore, condensin is involved in MATa-specific switching (Li et al. PLoS Genet 2019, 10.1371/journal.pgen.1008339), and condensin remains active on chromatin in cdc15 arrested cells, as shown on chr. XII (Lazar-Stefanita et al. EMBO J. 2017 10.15252/embj.201797342), which calls for determining the impact contribution of condensin in the recoil of the right ch.XII arm (Fig 4c) and on MAT switching.

There are several points here:

- Is the effect of cohesin observed in the MAT system specific to telophase or is it true in other cell-cycle phases?

Our new results show that cohesin depletion does not affect MAT switching when four different strains with all auxin system combinations are compared in the presence of auxin. Previously, when we compared minus versus plus auxin, we noticed diminished HO cutting efficiency. Therefore, we repeated the experiment using four isogenic strains (*SMC3*, *SMC3-aid**, *SMC3 + OsTIR1*, and *SMC3-aid* + OsTIR1*), which differ in their response to auxin and ability to degrade cohesin. This allowed us to compare all strains in the presence of auxin. As a result, we can now confirm that cohesin depletion does not affect MAT switching (see the new Figures 4b–d). Therefore, HR appears efficient after cohesin depletion. In agreement, the new ChIPs we have performed do not detect an increment in local cohesin after the HO DSB in telophase (but it does in cells arrested in G2/M).

- What is the mechanism behind this requirement (one may expect it not to depend on the sister since the HML donor is available within the damaged chromatid)?

As just said, we have changed our previous conclusion on cohesin and MAT switching. It was an effect of auxin addition rather than cohesin depletion.

- Does cohesin re-accumulate around the DSB site or genome-wide?

We have performed ChIP around the *HOcs*. We have found that it does accumulate in G2/M after HO induction, but it does not in telophase (new Figures 2c and S3). As for the global binding of cohesin, our chromatin fractionation data suggest there is ~2-fold increase in Smc1-Smc3, which also binds to the newly formed Scc1, rendering an overall increase in the chromatin-bound canonical complex (new Figures 2b and S2). Altogether, this suggests a genome-wide binding but with little role in the repair of HO DSBs.

- How does the Esp1 activity decay from anaphase onset?

We have not checked this here but it is an interesting question for a follow-up story.

- Is cohesin required for the horseshoe folding of chr. III involved in MATa-to-alpha switching?

Probably not in view of our new data in Figures 2c and 4b-d. The Piazza papers are cited and discussed.

- Contribution of condensin in the recoil of the right ch.XII arm (Fig 4c) and on MAT switching.

The role of condensin, which overtakes some cohesin function in late-M as the reviewer reminds, is worth studying indeed. However, we feel this deserves a separate and focus-on study. We does discuss, though, that condensin loading onto the arms in anaphase may prevent Smc1-Smc3 from loading after DSBs.

Other points:(4) Is the retrograde behavior in Fig. 4c dependent on recombination?

No, this is something we addressed in our previous paper (see Figure 4 in Nat Commun. 2019; 10(1):2862. doi: 10.1038/s41467-019-10742-8).

(5) Fig 3c: add a scheme of the system.

A scheme was already shown in the old Figure 2a (note that the old Fig 3c is now Fig S6).

(6) Fig 3b: annotate as in Fig 2b.

We have fixed this (now the referred figures are S6a and 3d, respectively).

(7) Authors used IAA concentrations 4- to 8-fold higher than commonly used. Given the solubility of IAA in DMSO (the most commonly used solvent), it is likely that authors treated their cells with >2% DMSO. This is expected to have broad transcriptional and physiological effects on yeast. A comparison of +IAA samples with a mock (DMSO) treatment would be more appropriate than a lack of treatment.

The IAA stock solution was 500 mM in DMSO, so the final DMSO concentration for an 8 mM IAA solution was 1.6% (v/v). Although the stock concentration was high and some precipitation was observed during preparation, we always heated, sonicated, and vigorously vortexed the stock tube before adding IAA to the cultures. Thus, we kept the uncertainty in the final IAA concentration to a minimum.